# Low Heat Availability Could Limit the Potential Spread of the Emerald Ash Borer to Northern Europe (Prognosis Based on Growing Degree Days per Year)

**DOI:** 10.3390/insects13010052

**Published:** 2022-01-02

**Authors:** Marina J. Orlova-Bienkowskaja, Andrzej O. Bieńkowski

**Affiliations:** A.N. Severtsov Institute of Ecology and Evolution, Russian Academy of Sciences, 119071 Moscow, Russia; bienkowski@yandex.ru

**Keywords:** emerald ash borer, EAB, Buprestidae, jewel beetles, Coleoptera, *Fraxinus*, forest pest

## Abstract

**Simple Summary:**

Emerald ash borer is a devastating pest of ash trees. This beetle, native to Asia and established in North America, European Russia and Ukraine is quickly spreading and approaching the borders of the European Union. We made the first prognosis of the potential range of this pest in Europe based on heat availability. Our calculations have shown that, in most European countries, the climate is warm enough for the establishment of the emerald ash borer. However, this pest would probably not be able to establish itself in some regions of Norway, Sweden, Finland, Ireland and Great Britain, because there is not enough heat to complete development (the summer is too cold and too short). Therefore, there is a hope that European ash (*Fraxinus excelsior*) could escape from the emerald ash borer in some parts of the British Isles and Scandinavia.

**Abstract:**

Emerald ash borer *Agrilus planipennis* (Coleoptera: Buprestidae) is one of 20 priority quarantine pests of the European Union. It is native to Asia and is established in the USA, Canada, European Russia, and Ukraine. We made the first prognosis of the potential range of *A. planipennis* in Europe based on heat availability. Mean annual growing degree days base 10 °C (AGDD_10_) was calculated for each grid square (0.25° × 0.25° latitude x longitude degrees) on the Earth’s surface. Minimal AGDD_10_ recorded in the grid squares currently occupied by *A. planipennis* was 714° in Asia, 705° in North America, and 711° in European Russia. *Agrilus planipennis* has never been recorded in localities with AGDD_10_ below 700°. If the phenotypic plasticity would not allow this species to overcome this threshold, cold regions of Europe would probably not be invaded by *A. planipennis*. Thus, *Fraxinus excelsior* could potentially escape from *A. planipennis* in some regions of Norway, Sweden, Finland, Ireland, and Great Britain.

## 1. Introduction

Emerald ash borer *Agrilus planipennis* (Fairmaire, 1888) (Coleoptera: Buprestidae) is an alien pest of ash trees (*Fraxinus* spp.) in the USA, Canada, European Russia and Ukraine [1,2]. This beetle, native to East Asia [3], was first recorded in North America in 2002, in Michigan [4] and in Europe in 2003, in Moscow [5]. Since that time, the pest has been spreading over both continents and killing millions of ash trees. *Agrilus planipennis* is a major threat to ash trees in Europe; it is included in the list of 20 priority quarantine pests of the European Union (EU) [6,7]. Now, the border of the *A. planipennis* range is just 120 km from the border of the EU; *A. planipennis* will inevitably appear in the EU soon [8,9]. The most commonly infested ash trees in European Russia and Ukraine are the green ash, *Fraxinus pennsylvanica* Marsh. This ash species was introduced for landscape plantings from North America in the 20th century and is known to be highly susceptible to emerald ash borer [10]. All ash species native to Europe (*Fraxinus excelsior* L., *F. ornus* L. and *F. angustifolia* Vahl.) are susceptible to *A. planipennis* [11]. It is very important to assess the potential range of the species in Europe, since it is an essential part in planning measures to mitigate economic and ecological losses from the future outbreak.

Climate and distribution of susceptible host plants are critical factors considered in predicting the potential distributions of alien species including *A. planipennis* [12]. Several prognoses of the potential range in Europe based on different approaches have been made [13,14,15]. Since the main factor limiting *A. planipennis* distribution is host availability, a high-resolution map of *A. planipennis* invasion risk for southern central Europe (Austria, Switzerland, Liechtenstein, southern Germany) was based on the distribution of ash trees *F. excelsior* [13]. Another model based on the maximum entropy modelling (MaxEnt) used the combination of 19 climatic parameters has shown that the regions of Europe adjacent to the known range of *A. planipennis* are suitable for this species; therefore, the climatic factors would not prevent the spread to adjacent regions [14]. The prognosis based on the minimum winter temperature as a possible limiting factor has shown that low winter temperatures would not prevent the spread of *A. planipennis* in all regions of Europe, except some eastern districts of European Russia [15].

Heat availability is one of the main factors determining the distribution of insect species [16]. The distribution of insects to the cold regions is often limited by the amount of heat (degree days) necessary to complete the life cycle. For example, the distribution of the two spotted oak buprestid, *Agrilus biguttatus* Fabricus, in Europe is limited by this factor [17]. It was hypothesized that low heat availability could limit the spread of *A. planipennis* to the north [18]. The only prognosis of *A. planipennis* potential range taking into account heat availability was recently made for Great Britain [19]. 

Here, we present the first prognosis of *A. planipennis* potential spread in the whole of Europe based on heat availability. Our calculations indicate that this climatic parameter could limit the potential spread of *A. planipennis* to Northern Europe.

## 2. Materials and Methods

### 2.1. Occurrences of Agrilus planipennis

We compiled a table of occurrences of *A. planipennis* in Asia, North America and Europe (Appendix A) using the information from current databases and articles [1,2,3,20,21]. The new occurrence in the city of Azov was added to the table (Appendix A). We detected characteristic exit holes and larval galleries and collected one dead adult *A. planipennis* from under the bark of *Fraxinus pennsylvanica* in the city of Azov on 8 September 2021. It is the first record of *A. planipennis* in the Rostov Region and the most south-western locality of the current range.

### 2.2. Calculation of Annual Growing Degree Days

The data on air temperature for each day from the beginning of 2003 through the end of 2020 were obtained from the ERA5-Land Global Atmospheric Reanalysis dataset for each grid square (0.25° × 0.25° latitude × longitude degrees) on the Earth’s surface [22]. We have chosen these last 18 years because *A. planipennis* was first recorded in Europe in 2003 [5]. The year of the first record always differs from the year of the establishment of the population. We can never know exactly when *A. planipennis* was established in the particular locality. Mean AGDD of an 18 year is an integrated parameter which gives us the general information about average heat availability in the particular locality.

The life cycle of *A. planipennis* in the colder regions (in particular, in Moscow) is 2 years, i.e., the larvae develop under the bark all year-round [23]. Therefore, the annual growing degree days (AGDD) were chosen as a parameter for the prognosis of the potential range. The base temperature of 10 °C was chosen because it is a standard base temperature previously used in the studies on *A. planipennis* phenology [24].

The mean AGDD_10_ were calculated in each grid square as follows: The daily mean temperature was calculated for each day as a mean of the temperatures at 00:00, 02:00, 04:00, 06:00, 08:00, 10:00, 12:00, 14:00, 16:00, 18:00, 20:00 and 22:00 UTC.Then the growing degree days base 10 °C from 1 January to 31 December were calculated for each year since 2003 to 2020 (AGDD_10_ in each particular year).Then the mean AGDD_10_ per year in 2003–2020 was calculated.

A detailed description of the calculation method and the computer code used is provided in the Appendix A. The results of the calculations are provided in the Excel table (Appendix A). 

### 2.3. Analysis and Visualisation of Results

The analysis of AGDD_10_ in each grid square occupied by *A. planipennis* in Asia, North America, European Russia and Ukraine allowed us to determine the minimum AGDD_10_ recorded in the occupied grid squares. Then we made a map of AGDD_10_ distribution in Europe and revealed the regions of Europe where AGDD_10_ is less than this threshold. The visualization was created using DIVA-GIS 7.5 [25]. The source of information for the *F. excelsior* range was the chorological maps for the main European woody species [26].

Shape files of countries and administrative units were obtained from DIVA GIS Free Spatial Data [27]. Shape files of the *F. excelsior* range, countries and administrative units were published in open-access sources published under a creative commons license [26,27]. 

## 3. Results

### 3.1. Heat Availability in the Regions Currently Occupied by A. planipennis

AGDD_10_ in the grid squares currently occupied by *A. planipennis* varies from 705 to 3676 (Table 1).

The distribution of AGDD_10_ in the current range of *A. planipennis* is shown in Figure 1, Figure 2 and Figure 3. The minimum values of AGDD_10_ recorded in Asia, North America and European Russia are very close to each other: slightly more than 700°. *Agrilus planipennis* has never been recorded in the grid squares with AGDD_10_ less than 700°.

The distribution of *A. planipennis* in its native range is not only limited by distribution of its host plants [12]. The host plant of *A. planipennis* in Asia, *Fraxinus mandshurica* Rupr., [28] occurs on the Sakhalin Island, some districts of Primorye Territory and Khabarovsk Territory, although *A. planipennis* is absent there [29] (Figure 4). The minimum heat availability (AGDD_10_) in grid squares occupied by *F. mandshurica* is 574°, while *A. planipennis* has been never recorded in the grid squares with AGDD_10_ below 700°. It indicates that heat availability could be one of the factors limiting the distribution of *A. planipennis* in the north of its range. 

### 3.2. Heat Availability in Europe

#### 3.2.1. General Distribution of Heat Availability

The map of the heat availability over the whole Europe is shown in Figure 5. In the majority of Europe, AGDD_10_ is above 700°. However, in the North (Norway, Sweden, Finland, Great Britain, Ireland, north of European Russia) and some mountain territories (the Carpathians, Pyrenees, Alps, Caucasus) there are regions where AGDD_10_ is below this threshold. 

Ash trees of different species are often planted outside their native ranges in Europe. Survival of these artificial plantings is important, but more important is the survival of ash trees in their native ranges, since ash trees play an important role in the communities of broad-leaved forests; hundreds of species of animals, plants and fungi ecologically depend on *Fraxinus* spp. [7]. The native range of *F. excelsior* occupies almost the whole of Europe, except the very south and very north (Figure 6). The heat availability is higher than 700° in the most part of the range, except some northern and mountain regions. 

#### 3.2.2. The Distribution of Heat Availability within the Range of Fraxinus Excelsior in Scandinavia

The only native ash species in the northern part of Europe is European ash (*F.*
*excelsior*) [26]. Heat availability is below 700° AGDD_10_ in the most part of Norway, Sweden, and Finland (Figure 7). There is a hope that these regions could potentially become the refuges for *F. excelsior**,* where ash trees could escape from *A. planipennis*. In Denmark, Estonia, southern regions of Sweden and regions along the southern coasts of Norway and Finland, AGDD_10_ is from 700 to 800 °C, i.e., about the minimum recorded in the territories occupied by *A. planipennis*.

#### 3.2.3. The Distribution of Heat Availability within the Range of Fraxinus Excelsior in British Isles

*Fraxinus excelsior* is common over the whole British Isles [26]. Our calculations have shown that AGDD_10_ in Ireland, Scotland, Northern Ireland, the most part of Wales and the northern half of England is less than 700° (Figure 8). 

## 4. Discussion

### 4.1. Heat Availability as a Limiting Factor of A. planipennis Range

Our data indicate that low heat availability could potentially limit the future spread of *A. planipennis* in Northern Europe. This conclusion is in accordance the conclusion by Webb et al. for the British Isles [19]. These authors calculated that northern half of the British Isles is unlikely to provide a suitable environment for *A. planipennis* to establish, because the GDD_10_ threshold of the start of the emergence of adults (230° GDD_10_ accumulated from 1 January) and the threshold of the peak of adult emergence (500° GDD_10_) are not met within a calendar year or are met in autumn [19].

It seems that the potential distribution of *A. planipennis* in Europe is similar to the current distribution of the two-spotted oak buprestid *Agrilus biguttatus* Fabricus. *Agrilus biguttatus* is widespread throughout Europe but reaches its northern-most limit in southern Sweden and in the northern half of the UK [16]. The host plants of *A. biguttatus* (*Quercus* spp.) are usual over the whole British Isles and in Scandinavia. However, the distribution of *A. biguttatus* is thermally limited with heat availability likely to be restrictive, rather than lethal summer or winter temperatures [16]. 

Prognoses that predict environmental suitability based on current location of cases for a pest that is still expanding into new regions may underestimate the area over which successful establishment is possible. However, our prognosis is based not only on the occurrences in invasive ranges in America and Europe, which are still expanding, but also on the occurrences in the native range in Asia, which is not expanding. The minimum AGDD_10_ in all three continents is almost the same (about 700°). Therefore, it seems that this figure is close to the minimum heat availability necessary for *A. planipennis* establishment. It is known that adult female *A. planipennis* body size (length and mass) depends on heat availability. The colder is the region, the smaller are adult females [37]. This phenotypic plasticity allows *A. planipennis* to survive in the colder regions. However, smaller *A. planipennis* females produce fewer eggs [37]. Obviously, the limit of this phenotypic plasticity exists: if the heat availability is too low, females are not able to survive or produce enough eggs for the establishment of the population 

### 4.2. Comparison of Low Heat Availability and Minimum Winter Temperature as Limiting Factors of A. planipennis Potential Distribution

It is interesting that low winter temperature and low heat availability pose different limitations to the potential range of *A. planipennis* in Europe (Figure 9).

Our previous calculations have shown *A. planipennis* populations survive in the territories where the minimum daily temperature is above 34° C, but do not survive in the territories where the daily temperature is below 34° C [15]. Temperature below 34° C is not rare in the east of European Russia but is extremely rare in Western Europe. Therefore, winter cold could limit of *A. planipennis* spread in the east of European Russia but could probably not limit its spread in Western Europe. Our current calculations have shown that fortunately the spread of the pest in the north of Europe could be potentially limited by low heat availability.

### 4.3. Perspectives for Future Researches

Variation in frequency and intensity of surveillance for *A. planipennis* between regions means that, for some areas, we can be less confident that a lack of report means that the pest is absent. The surveys of ash trees in the northern regions of European Russia and in the Far East are necessary for obtaining more detailed information about distribution of *A. planipennis* in the North.

AGDD_10_ in the high-altitude regions of the Alps, Pyrenees, Carpathians and Caucasus is less than 700°. The more detailed calculations of heat availability with higher resolution and the comparison with the high-resolution map of the distribution of ash trees (*Fraxinus excelsior*, *F. ornus* and *F. angustifolia*) are necessary to assess whether some high-altitude regions could become potential refuges of ash trees from *A. planipennis*.

Warmer temperatures appear to have contributed to recent northward range shifts of some native wood-boring forest pests in Europe, including jewel beetles (Coleoptera: Buprestidae): *Coraebus florentinus* Herbst and *Agrilus sulcicollis* Lacordaire [16]. Therefore, the future warming could potentially affect the spread of *A. planipennis* to the North. The analysis of the possible impact of the future climate change on the potential spread of *A. planipennis* is a perspective for future research. 

It is known that the duration of the life cycle of *A. planipennis* depends on the temperature: in the warmer regions, the life cycle is usually 1 year, while in the colder regions it is 2 years [23]. It is important to reveal in what regions of Europe the potential development could be 1 year, because in such regions the spread of *A. planipennis* could be faster than in others. The estimation of the potential duration of the life cycle in different parts of Europe is also a perspective for future research.

## 5. Conclusions

Minimum AGDD_10_ recorded in the localities of *A. planipennis* in the three continents are almost the same: 714° in Asia, 705° in North America, and in 711° Europe. *Agrilus plenipennis* has been never recorded in the regions with AGDD_10_ below 700°.In the majority of Europe, AGDD_10_ is more than 700°. Therefore, low heat availability would probably not limit the spread of *A. planipennis* in most of European countries.Heat availability in most regions of Norway, Sweden, Finland, and Ireland and in the northern half of Great Britain is less than 700°. If the phenotypic plasticity would not allow the pest to overcome this threshold, *A. planipennis* would potentially not establish in these regions. Therefore, *Fraxinus excelsior* could potentially escape from *A. planipennis* in these regions.

## Figures and Tables

**Figure 1 insects-13-00052-f001:**
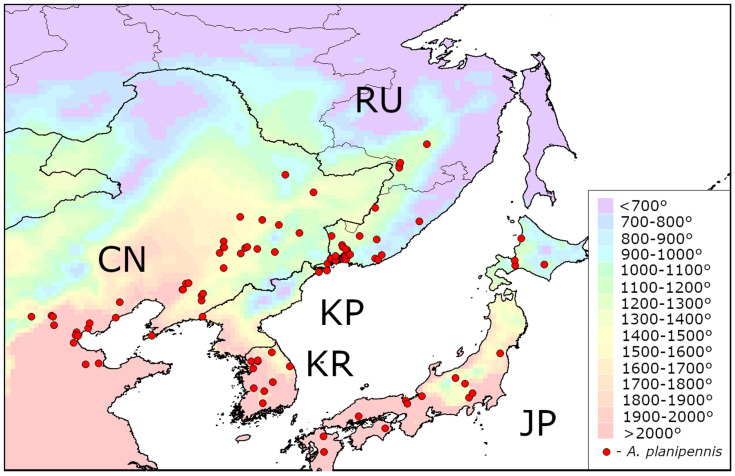
Heat availability in the native range of *Agrilus. planipennis* in Asia. Mean AGDD_10_ per year in 2003–2020 is indicated with colours (see the legend). CN—China, JP—Japan, KP—Democratic People’s Republic of Korea, KR—Republic of Korea, RU—Russia. The information about occurrences of *A. planipennis* was obtained from the review of its native range [3].

**Figure 2 insects-13-00052-f002:**
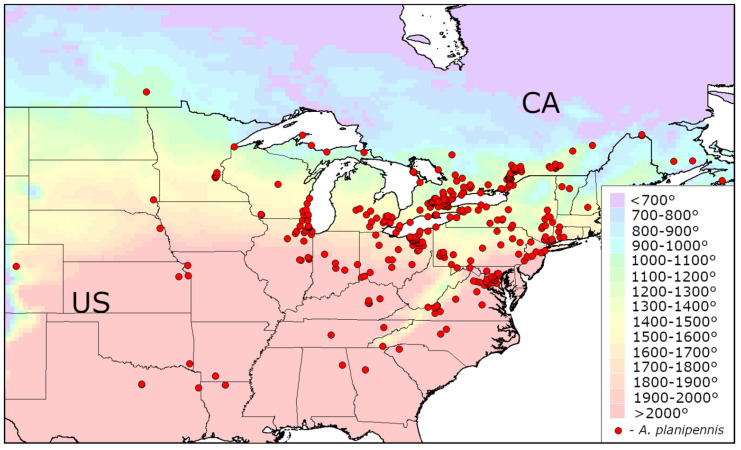
Heat availability in the current invasive range of *Agrilus. planipennis* in North America. Mean AGDD_10_ per year in 2003–2020 is indicated with colours (see the legend). CA—Canada, US—United States. The information about occurrences of *Agrilus planipennis* was obtained from Emerald Ash Borer Info and Global Biodiversity Information Facility [1,21].

**Figure 3 insects-13-00052-f003:**
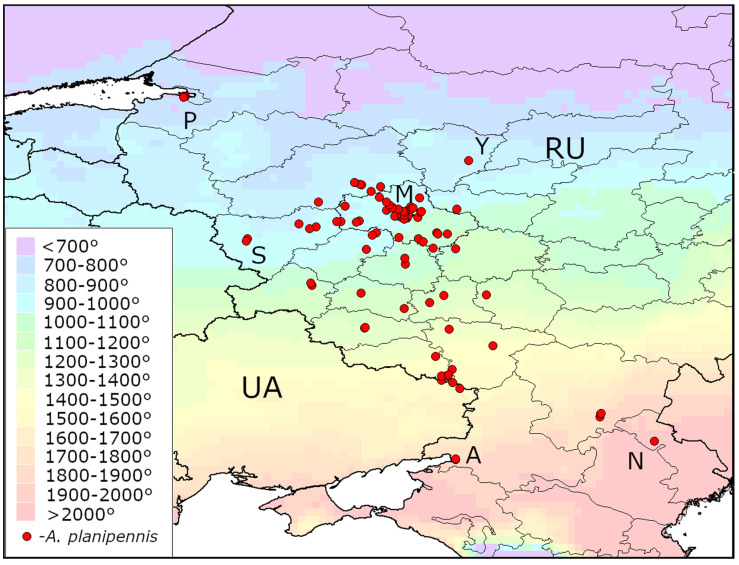
Heat availability in the current range of *Agrilus planipennis* in European Russia and Ukraine. Mean AGDD_10_ per year in 2003–2020 is indicated with colours (see the legend). RU—Russia, UA—Ukraine. The information about occurrences of *A. planipennis* was obtained from recent publications [9,19]. Some occurrences are indicated with letters: A—Azov, M—Moscow, P—Saint Petersburg, S—Smolensk, Y—Yaroslavl, N—Nikolskoe Village (Astrakhan Region). The new occurrence in Azov is published for the first time.

**Figure 4 insects-13-00052-f004:**
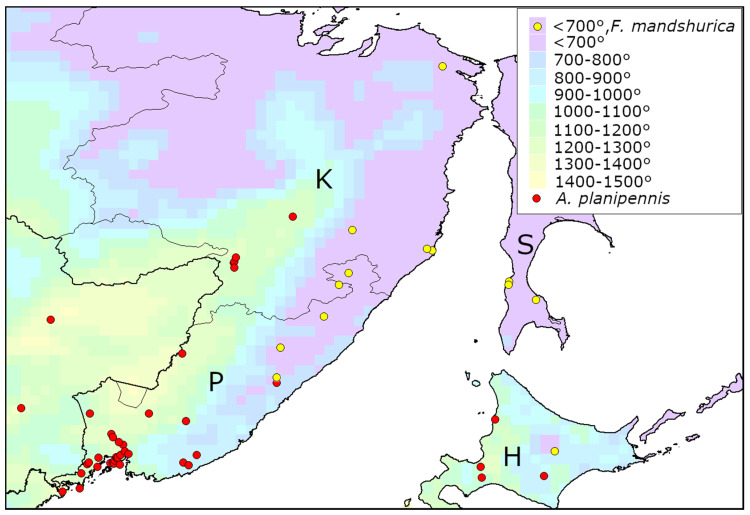
Distribution of *Agrilus planipennis* and its host plant *Fraxinus mandshurica* in the northern part of *A. planipennis* native range in Asia. Mean AGDD_10_ per year in 2003–2020 is indicated with colours (see the legend). Sources of the information on occurrences: *A. planipennis*—recent review of the native range [3], *Fraxinus mandshurica*—Global Biodiversity Information Facility [29]. Only localities of *F. mandshurica* in the grid squares with AGDD_10_ less than 700° are shown.

**Figure 5 insects-13-00052-f005:**
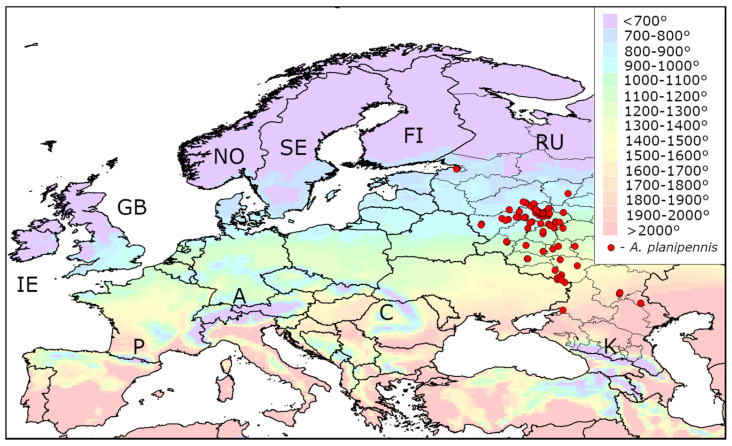
Heat availability in Europe. Mean AGDD_10_ per year in 2003-2020 is indicated with colours (see the legend). The information about occurrences of *Agrilus planipennis* was obtained from recent publications [9,19]. IE—Ireland, GB—Great Britain, NO—Norway, SE—Sweden, FI—Finland, RU—Russia, A—Alps, C—Carpathians, K—Caucasus, P—Pyrenees.

**Figure 6 insects-13-00052-f006:**
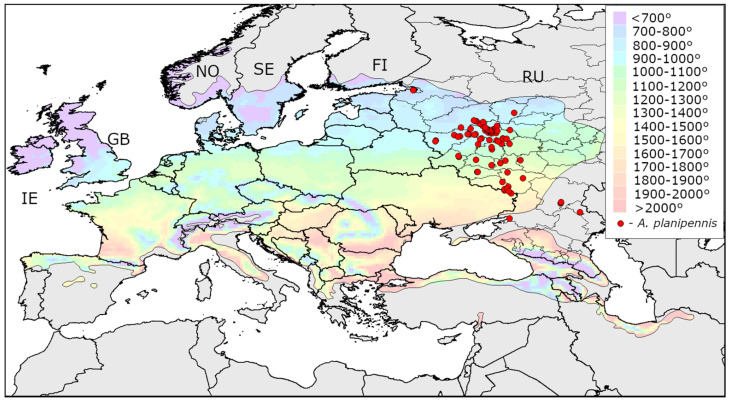
Heat availability in the native range of *Fraxinus excelsior* in Europe. Mean AGDD_10_ per year in 2003–2020 within the native range of *F. excelsior* is indicated with colours (see the legend). FI—Finland, GB—Great Britain, IE—Ireland, NO—Norway, RU—Russia, SE—Sweden.

**Figure 7 insects-13-00052-f007:**
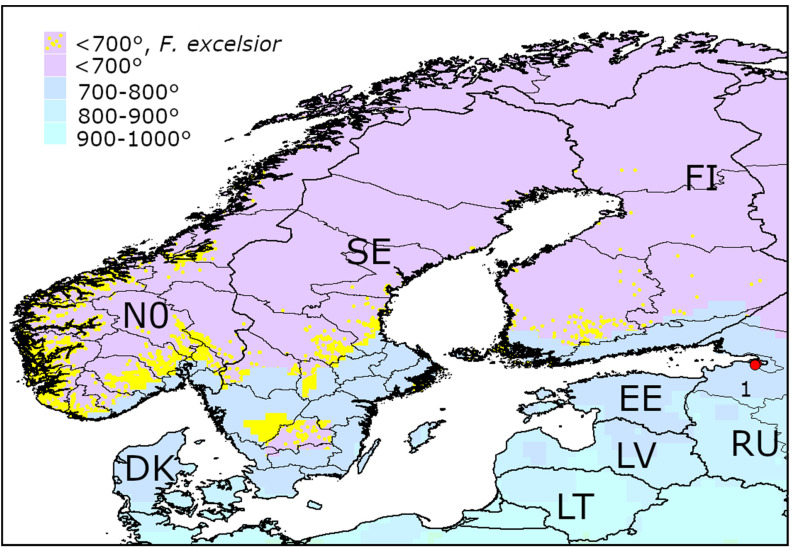
Occurrences of *Fraxinus excelsior* in the territories with AGDD_10_ below 700° in Northern Europe. Mean AGDD_10_ per year in 2003–2020 is indicated with colours (see the legend). DK—Denmark, EE—Estonia, FI—Finland, LT—Lithuania, LV—Latvia, NO—Norway, RU—Russia, SE—Sweden. 1—occurrences of *Agrilus planipennis* in Saint Petersburg. The information about the occurrences of *F. excelsior* was obtained from Global Biodiversity Information Facility [30,31,32].

**Figure 8 insects-13-00052-f008:**
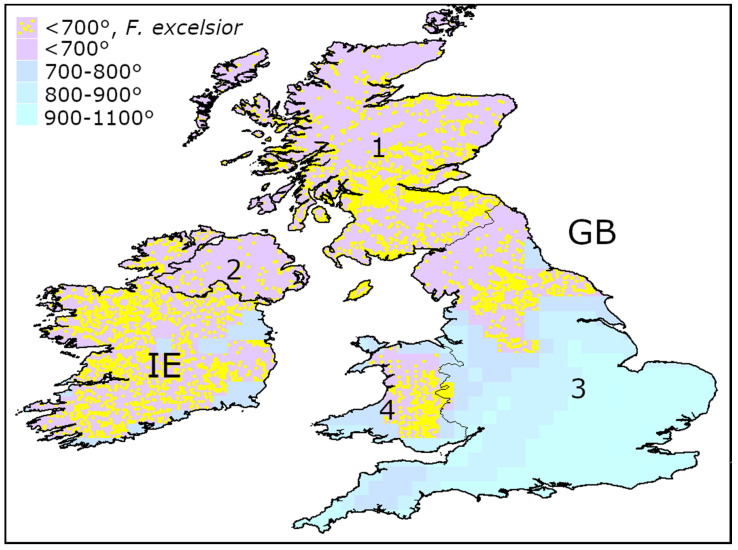
Occurrences of *Fraxinus excelsior* in the territories with AGDD_10_ below 700° in British Isles. GB—Great Britain, IE—Ireland. 1—Scotland, 2—Northern Ireland, 3—England, 4—Wales. Mean AGDD_10_ per year in 2003–2020 is indicated with colours (see the legend). The information about the occurrences of *F. excelsior* was obtained from Global Biodiversity Information Facility [33,34,35,36].

**Figure 9 insects-13-00052-f009:**
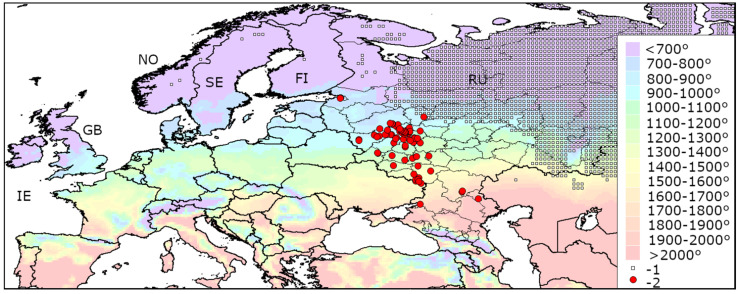
Distribution minimum winter temperature and heat availability in Europe. Mean AGDD_10_ per year in 2003–2020 is indicated with colours (see the legend). 1—territories, which are not suitable for *Agrilus planipennis* establishment because of the extreme winter cold [15], 2—localities of *A. planipennis*. FI—Finland, GB—Great Britain, IE—Ireland, NO—Norway, RU—Russia, SE—Sweden.

**Table 1 insects-13-00052-t001:** Minimum and maximum mean AGDD_10_ in 2003–2020 (°C) in the grid squares currently occupied by *A. planipennis* in different continents.

Continent	Minimum	Maximum
Asia	714	2778
North America	705	3676
Europe (Russia and Ukraine)	711	2046

## Data Availability

All data generated during this study are in the electronic supplements (Appendix A).

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
