# Peer review of "Low Heat Availability Could Limit the Potential Spread of the Emerald Ash Borer to Northern Europe (Prognosis Based on Growing Degree Days per Year)"

_insects, 2022, doi:10.3390/insects13010052_

Round 1
Reviewer 1 Report
This paper consists of three steps:
- Estimation of the minimum annual growing degree days above a base of 10oC (AGDD10) required for Emerald Ash Borer (EAB) to breed successfully based on all reported locations of infestations of EAB available in the literature.
- Presentation of heatmap plots for Europe, Asia and North America overlaid with all reported locations of EAB infestations.
- Presentation of host specific heatmap plots for Europe and Asia.
The analysis is very basic with no attempt to the impact of AGDD on the duration of the lifecycle or statistical analysis of the data. The paper does not present either a statistical or mathematical model but rather provides evidence for a potential predictor of the geographic limits of EAB spread.
The analysis presented in the paper could be strengthened in a number of ways:
- Stability of estimated AGDD: I suggest that the authors calculate and plot the estimated minimum AGDD as a function of time (i.e. what would the estimate have been if you had carried out this analysis in say 2003). This would enable exploration of the stability of the estimated minimum AGDD over time. Information on the stability of the estimate would allow consideration of whether the absence of the pest from areas with AGDD less than the proposed threshold is an intrinsic property of the pest or whether it may be due to lack of opportunity for the pest to establish. It is important to acknowledge that models which predict environmental suitability based on current location of cases for a pest or pathogen that is still expanding into new regions may underestimate the area over which successful establishment is possible.
- Impact of annual variation in AGDD: The use in the paper of an 18-year average means that information on the variation in AGDD in each grid cell is lost. This is important both is estimating the minimum AGDD for invasion by EAB and, if assuming an AGDD threshold exists, the potential areas which the pest could establish. Is the estimated minimum AGDD required higher if you plot the AGDD for the year of first reported case in each grid cell? Might annual variation in temperatures facilitate spread into new regions?
- One versus two-year lifecycle: This complicating feature has been glossed over by the authors however it is a clear component in how the pest can adapt to cooler regions. Is there sufficient data on the duration of the lifecycle at each data point to calculate the minimum number of AGDD for regions in which the lifecycle has been reported as 1 year, mixed 1 and 2 year, and 2 year only (the authors have another paper in which they gathered some of this information – can this be updated?).
- Missing data: The paper is missing discussion of the quantity and quality of the data used in the analysis. In particular, the data for distribution of mandushurica/ manjurica appears, from Figure 5, to be recorded presence rather than definite absence of the species and thus may not fully map the distribution of the host. Similarly, the host data for the F. excelsior, does not show areas within the boundaries where it is absent (such as some mountainous areas). The authors should discuss potential impact of data quality on their conclusions. The variation in frequency and intensity of surveillance for EAB between regions mean that for some areas we can be less confident that a lack of report means that the pest is absent.
- Relevance of climate in mountainous areas: Many of these maybe above the tree line.
- Combining AGDD with other potential predictors of establishment: Can this analysis be combined or directly compared with previous analysis by the authors on the role of minimal winter temperatures Orlova-Bienkowskaja et al (2020), by Webb et al (2021) and the MaxEnt model predictions by Flø et al (2015) to see whether the proposed use of AGDD gives different bounds to the potential spread of EAB. What does calculation of AGDD add to these other measures?
Specific points
- Supplementary S2 (code) was not supplied
- The authors should remove references to the work as a ‘model’ – there is no model presented in the paper.
- Line 78-9: what is the AGDD at this data point, specifically is it below the proposed threshold?
- The paper requires some restructuring. As it is currently presented, the discussion contains additional description of methods and new results. The methods section does not contain sufficient information how the data is analysed, and the precise source of data used for each host range. Figures 5-10 should be moved to the results.
- The first paragraph of the discussion is weak – obviously the distribution of the pest is limited by host availability. The second sentence is backed-up by a self-citation – please use another source. The discussion also lacks a comparison of the results with other papers that consider the potential range of EAB.
- Consider presenting data on the % of all grid squares occupied by F. ornus that have AGDD < 700.
- Conclusion point 3: here need to mention dual threat of ‘chalara’ ash dieback
Grammatical suggestions
- Line 87: delete ‘initial’
- Line 107: replace minimal with minimum
- Line 111: replace ‘have been’ with ‘were’
- Line 121-122: should be “never been”
- Line 143-144: delete final sentence of this paragraph as does not add anything.
- Line 169: replace “than” with “when”
- Line 170: replace “the” with “a” and delete “about start at a”
- Line 171: insert “a” i.e. “correspond to a threshold of …”
- Figure numbers: check figure numbers - numbering jumps from 6 to 9.
- Line 215-217: this sentence does not make grammatical sense
- Line 225: Delete “According to our model” – this data came from ERA-5 not your model
- Line 228: either delete or reword “so we are unable to make precise …”
- Line 246: The minimum (18-year) average AGDD …
Line 260: replace “the low” with “low”
Author Response
- Stability of estimated AGDD: I suggest that the authors calculate and plot the estimated minimum AGDD as a function of time (i.e. what would the estimate have been if you had carried out this analysis in say 2003). This would enable exploration of the stability of the estimated minimum AGDD over time. Information on the stability of the estimate would allow consideration of whether the absence of the pest from areas with AGDD less than the proposed threshold is an intrinsic property of the pest or whether it may be due to lack of opportunity for the pest to establish.
- We have calculated AGDD for each grid square on the Earth in each year 2003-2020. All these AGDD are shown in Table S3. We have 18 AGDD for each of 1038240 grids. We have chosen the mean AGDD as an integrating parameter which characterize the heat availability in each grid square. The minimal AGDD recorded in the particular grid square is less suitable than the mean AGDD, because the life cycle is often 2-year. For example, if the summer 2008 was cold in the particular grid square (AGDD was below the threshold), but the summer 2009 was warm, the population of the pest can survive.
- It is important to acknowledge that models which predict environmental suitability based on current location of cases for a pest or pathogen that is still expanding into new regions may underestimate the area over which successful establishment is possible.
- We added the discussion of this issue to the Discussion.
- Impact of annual variation in AGDD:The use in the paper of an 18-year average means that information on the variation in AGDD in each grid cell is lost. This is important both is estimating the minimum AGDD for invasion by EAB and, if assuming an AGDD threshold exists, the potential areas which the pest could establish. Is the estimated minimum AGDD required higher if you plot the AGDD for the year of first reported case in each grid cell? Might annual variation in temperatures facilitate spread into new regions?
- Thank you for this comment. We corrected “Materials and Methods” to make our idea clearer. The year of the first record always differs from the year of the establishment of the population. We never know when the population was established and, therefore, we are unable to determine what was the AGDD in this year. Dendrochronological studies show that 10-15 years could pass between the establishment of the population and the first finding of A. planipennis. Mean AGDD of an 18 year is an integrated parameter which gives us the general information about average heat availability.
- One versus two-year lifecycle: This complicating feature has been glossed over by the authors however it is a clear component in how the pest can adapt to cooler regions. Is there sufficient data on the duration of the lifecycle at each data point to calculate the minimum number of AGDD for regions in which the lifecycle has been reported as 1 year, mixed 1 and 2 year, and 2 year only (the authors have another paper in which they gathered some of this information – can this be updated?).
- We agree that the duration of life cycle is a very important issue which affects the possibility of A. planipennis to establish. Unfortunately, the information about the life cycle is scarce. The life cycle was studied in Europe only once in only one locality (our study in Moscow Region mentioned by you). Therefore, we have not enough information to answer your very good question.
- Missing data: The paper is missing discussion of the quantity and quality of the data used in the analysis. In particular, the data for distribution of mandushurica/ manjurica appears, from Figure 5, to be recorded presence rather than definite absence of the species and thus may not fully map the distribution of the host. Similarly, the host data for the F. excelsior, does not show areas within the boundaries where it is absent (such as some mountainous areas). The authors should discuss potential impact of data quality on their conclusions. The variation in frequency and intensity of surveillance for EAB between regions mean that for some areas we can be less confident that a lack of report means that the pest is absent.
- We added an explanation to the figure to make our idea clearer. We show only those occurrences of Fraxinus mandushurica that are situated in the regions with AGDD10 below 700. We did not show other occurrences of Fraxinus mandushurica, since they were not important for our conclusion. We just wanted to show that mandushurica can live in the regions with AGDD10 below 700.We excluded the analysis of the spread of A. planipennis to mountain areas. We added the discussion about the possible impact of intensity of surveillance on the confidence of lack of the reports.
- Relevance of climate in mountainous areas: Many of these maybe above the tree line.
- We decided to exclude the discussion of the mountain areas from this article.
- Combining AGDD with other potential predictors of establishment: Can this analysis be combined or directly compared with previous analysis by the authors on the role of minimal winter temperatures Orlova-Bienkowskaja et al (2020), by Webb et al (2021) and the MaxEnt model predictions by Flø et al (2015) to see whether the proposed use of AGDD gives different bounds to the potential spread of EAB. What does calculation of AGDD add to these other measures?
- Thank you. According to this recommendation we added the comparison of Comparison of two factors limiting the potential distribution: heat availability and minimum winter temperature as limiting factors of A. planipennis potential distribution into the “Discussion” section. The very good prediction of the potential distribution of A. planipennis in British Isles made by Webb et al (2021) is discussed in the section 4.1. Our data on British Isles are in accordance with the data by Webb et al. The MaxEnt model by Flø et al (2015) is mentioned in the introduction. This model is weak, because it is based on erroneous occurrence information. We criticized this model in our previous article (Orlova–Bienkowskaja, Bie ´nkowski, 2020) and decided not to repeat these arguments in the current article.
Specific points
Specific points
- Supplementary S2 (code) was not supplied
- Thank you. It is fixed.
- The authors should remove references to the work as a ‘model’ – there is no model presented in the paper.
- We removed the word “model”.
- Line 78-9: what is the AGDD at this data point, specifically is it below the proposed threshold?
- Agrilus planipennis was recorded in Xinjiang Uygur Autonomous Region only once, in the review article [Wei X, Reardon D, Yun W, Sun JH (2004) Emerald ash borer, Agrilus planipennis Fairmaire (Coleoptera: Buprestidae), in China: a review and distribution survey. Acta Entomol Sinica 47:679–685]. It seems that this record was a mistake, since this record was not mentioned in the subsequent articles on A. planipennis distribution in China. We deleted the mention of this record from our manuscript, since the particular locality was not indicated in the article by Wei et al.
- The paper requires some restructuring. As it is currently presented, the discussion contains additional description of methods and new results. The methods section does not contain sufficient information how the data is analysed, and the precise source of data used for each host range. Figures 5-10 should be moved to the results.
- We have restructured the paper and transferred figures to results. The source of the host ranges is indicated in Methods. It is chorological maps for the main European woody species. The reference is provided.
We decided to exclude the discussion of Fraxinus ornus and F. angustifolia from the article, since they do not occur in the north, while the article is about potential spread in Northern Europe.
- The first paragraph of the discussion is weak – obviously the distribution of the pest is limited by host availability. The second sentence is backed-up by a self-citation – please use another source. The discussion also lacks a comparison of the results with other papers that consider the potential range of EAB.
- We deleted the first paragraph of the discussion and added comparison with the results of other papers consider potential range of EAB.
- Consider presenting data on the % of all grid squares occupied by F. ornus that have AGDD < 700.
- We excluded F. ornus from the paper.
- Conclusion point 3: here need to mention dual threat of ‘chalara’ ash dieback
- We deleted the last sentence of the point 3. : “Therefore, Fraxinus excelsior could potentially escape from A. planipennis in these regions.”
Grammatical suggestions
- Line 87: delete ‘initial’
- OK
- Line 107: replace minimal with minimum
- OK
- Line 111: replace ‘have been’ with ‘were’
- OK
- Line 121-122: should be “never been”
- OK
- Line 143-144: delete final sentence of this paragraph as does not add anything.
- OK
- Line 169: replace “than” with “when”
- OK
- Line 170: replace “the” with “a” and delete “about start at a”
- OK
- Line 171: insert “a” i.e. “correspond to a threshold of …”
- OK
- Figure numbers: check figure numbers - numbering jumps from 6 to 9.
- It is fixed.
- Line 215-217: this sentence does not make grammatical sense
- It is reworded.
- Line 225: Delete “According to our model” – this data came from ERA-5 not your model
- Yes.
- Line 228: either delete or reword “so we are unable to make precise …”
- Yes.
- Line 260: replace “the low” with “low”
- OK
Reviewer 2 Report
Considering the potential threat of the emerald ash borer invasion to Europe, this article is timely. I have no major issues except some editorial comments.
2) Line 25 - please give the grid unit (25 x 25) - square kilometers or latitude x longitude degrees?
3) Lines 46-48. Please rewrite these too sentences or combined them. I noted the suggestion in the manuscript - "Most commonly infested ash trees in ,,,, are the green ash, Fraxinus pensylvanica Marsh. This ash species was introduced for landscape plantings from North America in ???? and is known to be highly susceptible to emerald ahs borer (11).
4) Lines 152 - 154. Two most recently published articles may be relevant to the discussion of factors liming the emerald ash borer distribution. I strongly suggest the authors take a look at these publications to see if there are relevance there"
Dang et al 2020: Journal of Pest Science
https://doi.org/10.1007/s10340-020-01308-5x
Duan et al. 2020: Journal of Economic Entomology, 113(3), 2020, 1145–1151: https://doi.org/10.1093/jee/toaa048
Duan et al. 2020: Journal of Economic Entomology, 114(1), 2021, 201–208
https://doi.org/10.1093/jee/toaa252
Author Response
Thank you for the review of our manuscript.
- Line 25 - please give the grid unit (25 x 25) - square kilometers or latitude x longitude degrees?
- Yes. It is added. (latitude x longitude degrees).
2) Lines 46-48. Please rewrite these too sentences or combined them. I noted the suggestion in the manuscript - "Most commonly infested ash trees in ,,,, are the green ash, Fraxinus pensylvanica Marsh. This ash species was introduced for landscape plantings from North America in ???? and is known to be highly susceptible to emerald ahs borer (11
- Thank you. the sentences are rewritten.
3) Lines 152 - 154. Two most recently published articles may be relevant to the discussion of factors limiting the emerald ash borer distribution. I strongly suggest the authors take a look at these publications to see if there are relevance there"
Dang et al 2020: Journal of Pest Science
https://doi.org/10.1007/s10340-020-01308-5x
Duan et al. 2020: Journal of Economic Entomology, 113(3), 2020, 1145–1151: https://doi.org/10.1093/jee/toaa048
Duan et al. 2020: Journal of Economic Entomology, 114(1), 2021, 201–208
https://doi.org/10.1093/jee/toaa252
- Thank you! The first of these publications is relevant to our work. We used it and included into the reference list. Two other articles are devoted to the winter temperatures and diapause. I will use them in my next article.
Round 2
Reviewer 1 Report
I am satisfied that the authors have responded to the original comments and made significant improvements to the paper.